# COMPUTATION REALLOCATION FOR OBJECT DETECTION

**Feng Liang**[1]**, Chen Lin**[1]**, Ronghao Guo**[1]**, Ming Sun**[1]**, Wei Wu**[1]**, Junjie Yan**[1]**, Wanli Ouyang**[2]
[1]Sensetime Research Group
{liangfeng,linchen,guoronghao,sunming1,wuwei,yanjunjie}@sensetime.com
[2]The University of Sydney
wanli.ouyang@sydney.edu.au

## ABSTRACT

The allocation of computation resources in the backbone is a crucial issue in object detection. However, classification allocation pattern is usually adopted directly to object detector, which is proved to be sub-optimal. In order to reallocate the engaged computation resources in a more efficient way, we present CR-NAS (Computation Reallocation Neural Architecture Search) that can learn computation reallocation strategies across different feature resolution and spatial position diectly on the target detection dataset. A two-level reallocation space is proposed for both stage and spatial reallocation. A novel hierarchical search procedure is adopted to cope with the complex search space. We apply CR-NAS to multiple backbones and achieve consistent improvements. Our CR-ResNet50 and CR-MobileNetV2 outperforms the baseline by 1.9% and 1.7% COCO AP respectively without any additional computation budget. The models discovered by CR-NAS can be equiped to other powerful detection neck/head and be easily transferred to other dataset, e.g. PASCAL VOC, and other vision tasks, e.g. instance segmentation. Our CR-NAS can be used as a plugin to improve the performance of various networks, which is demanding.

## 1 INTRODUCTION

Object detection is one of the fundamental tasks in computer vision. The backbone feature extractor is usually taken directly from classification literature (Girshick, 2015; Ren et al., 2015; Lin et al., 2017a; Lu et al., 2019). However, comparing with classification, object detection aims to know not only *what* but also *where* the object is. Directly taking the backbone of classification network for object detectors is sub-optimal, which has been observed in Li et al. (2018). To address this issue, there are many approaches either manually or automatically modify the backbone network. Chen et al. (2019) proposes a neural architecture search (NAS) framework for detection backbone to avoid expert efforts and design trails. However, previous works rely on the prior knowledge for classification task, either inheriting the backbone for classification, or designing search space similar to NAS on classification. This raises a natural question: *How to design an effective backbone dedicated to detection tasks?*

To answer this question, we first draw a link between the Effective Receptive Field (ERF) and the computation allocation of backbone. The ERF is only small Gaussian-like factor of theoretical receptive field (TRF), but it dominates the output (Luo et al., 2016). The ERF of image classification task can be easily fulfilled, *e.g.* the input size is 224×224 for the ImageNet data, while the ERF of object detection task need more capacities to handle scale variance across the instances, *e.g.* the input size is 800×1333 and the sizes of objects vary from 32 to 800 for the COCO dataset. Lin et al. (2017a) allocates objects of different scales into different feature resolutions to capture the appropriate ERF in each stage. Here we conduct an experiment to study the differences between the ERF of several FPN features. As shown in Figure 1, we notice the allocation of computation across different resolutions has a great impact on the ERF. Furthermore, appropriate computation allocation across spacial position (Dai et al., 2017) boost the performance of detector by affecting the ERF.

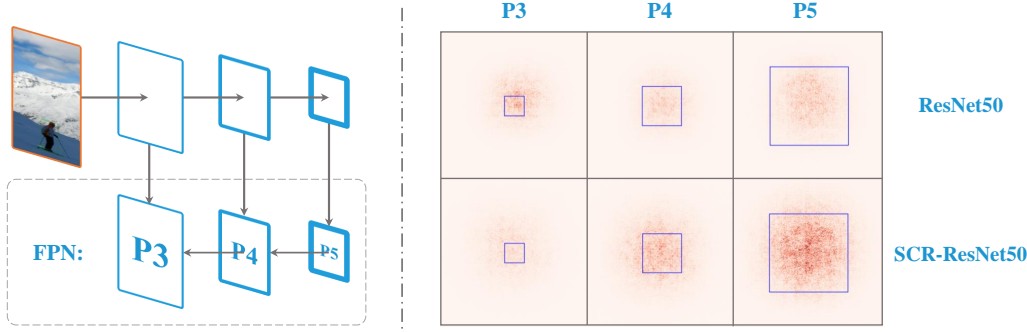

Figure 1: Following the instructions in Luo et al. (2016), we draw the ERF of FPN in different resolution features. The size of base plate is 512×512, with respective anchor boxes ({64, 128, 256} for $\{p_3, p_4, p_5\}$) drawn in. The classification CNNs ResNet50 tends to have redundant ERF for high resolution features $p_3$ and limited ERF for low resolution features $p_5$. After stage reallocation, our SCR-ResNet50 has more balanced ERF across all resolutions which leads to a high performance.

Based on the above observation, in this paper, we aim to automatically design the computation allocation of backbone for object detectors. Different from existing detection NAS works (Ghiasi et al., 2019; Ning Wang & Shen, 2019) which achieve accuracy improvement by introducing higher computation complexity, we reallocate the engaged computation cost in a more efficient way. We propose computation reallocation NAS (CR-NAS) to search the allocation strategy directly on the detection task. A two-level reallocation space is conducted to reallocate the computation across different resolution and spatial position. In stage level, we search for the best strategy to distribute the computation among different resolution. In operation level, we reallocate the computation by introducing a powerful search space designed specially for object detection. The details about search space can be found in Sec. 3.2. We propose a hierarchical search algorithm to cope with the complex search space. Typically in stage reallocation, we exploit a reusable search space to reduce stage-level searching cost and adapt different computational requirements.

Extensive experiments show the effectiveness of our approach. Our CR-NAS offers improvements for both fast mobile model and accurate model, such as ResNet (He et al., 2016), MobileNetV2 (Sandler et al., 2018), ResNeXt (Xie et al., 2017). On the COCO dataset, our CR-ResNet50 and CR-MobileNetV2 can achieve 38.3% and 33.9% AP, outperforming the baseline by 1.9% and 1.7% respectively without any additional computation budget. Furthermore, we transfer our CR-ResNet and CR-MobileNetV2 into the another ERF-sensitive task, instance segmentation, by using the Mask RCNN (He et al., 2017) framework. Our CR-ResNet50 and CR-MobileNetV2 yields 1.3% and 1.2% COCO segmentation AP improvement over baseline.

To summarize, the contributions of our paper are three-fold:

- We propose computation reallocation NAS(CR-NAS) to reallocate engaged computation resources. To our knowledge, we are the first to dig inside the computation allocation across different resolution.

- We develop a two-level reallocation space and hierarchical search paradigm to cope with the complex search space. Typically in stage reallocation, we exploit a reusable model to reduce stage-level searching cost and adapt different computational requirements.

- Our CR-NAS offers significant improvements for various types of networks. The discovered models show great transferablity over other detection neck/head, e.g. NAS-FPN (Cai & Vasconcelos, 2018), other dataset, e.g. PASCAL VOC (Everingham et al., 2015) and other vision tasks, e.g. instance segmentation (He et al., 2017).

## 2 RELATED WORK

**Neural Architecture Search(NAS)** Neural architecture search focus on automating the network architecture design which requires great expert knowledge and tremendous trails. Early NAS approaches (Zoph & Le, 2016; Zoph et al., 2018) are computational expensive due to the evaluating of each candidate. Recently, weight sharing strategy (Pham et al., 2018; Liu et al., 2018; Cai et al., 2018; Guo et al., 2019) is proposed to reduce searing cost. One-shot NAS method (Brock et al., 2017; Bender et al., 2018; Guo et al., 2019) build a directed acyclic graph $\mathcal{G}$ (a.k.a. supernet) to subsume all architectures in the search space and decouple the weights training with architectures searching. NAS works only search for operation in the certain layer. our work is different from them by searching for the computation allocation across different resolution. Computation allocation across feature resolutions is an obvious issue that has not been studied by NAS. We carefully design a search space that facilitates the use of existing search for finding good solution.

**NAS on object detection.** There are some work use NAS methods on object detection task (Chen et al., 2019; Ning Wang & Shen, 2019; Ghiasi et al., 2019). Ghiasi et al. (2019) search for scalable feature pyramid architectures and Ning Wang & Shen (2019) search for feature pyramid network and the prediction heads together by fixing the architecture of backbone CNN. These two work both introduce additional computation budget. The search space of Chen et al. (2019) is directly inherited from the classification task which is suboptimal for object detection task. Peng et al. (2019) search for dilated rate on channel level in the CNN backbone. These two approaches assume the fixed number of blocks in each resolution, while we search the number of blocks in each stage that is important for object detection and complementary to these approaches.

## 3 METHOD

### 3.1 BASIC SETTINGS

Our search method is based on the Faster RCNN (Ren et al., 2015) with FPN (Lin et al., 2017a) for its excellent performance. We only reallocate the computation within the backbone, while fix other components for fair comparison.

For more efficient search, we adopt the idea of one-shot NAS method (Brock et al., 2017; Bender et al., 2018; Guo et al., 2019). In one-shot NAS, a directed acyclic graph $\mathcal{G}$ (a.k.a. supernet) is built to subsume all architectures in the search space and is trained only once. Each architecture $g$ is a subgraph of $\mathcal{G}$ and can inherit weights from the trained supernet. For a specific subgraph $g \in \mathcal{G}$, its corresponding network can be denoted as $\mathcal{N}(g, w)$ with network weights $w$.

### 3.2 TWO-LEVEL ARCHITECTURE SEARCH SPACE

We propose Computation Reallocation NAS (CR-NAS) to distribute the computation resources in two dimensions: stage allocation in different resolution, convolution allocation in spatial position.

#### 3.2.1 STAGE REALLOCATION SPACE

The backbone aims to generate intermediate-level features $C$ with increasing downsampling rates $4\times$, $8\times$, $16\times$, and $32\times$, which can be regarded as 4 stages. The blocks in the same stage share the same spatial resolution. Note that the FLOPs of a single block in two adjacent spatial resolutions remain the same because a downsampling/pooling layer doubles the number of channels. So given the number of total blocks of a backbone $N$, we can reallocate the number of blocks for each stage while keep the total FLOPs the same. Figure 2 shows our stage reallocation space. In this search space, each stage contains several branches, and each branch has certain number of blocks. The numbers of blocks in different branches are different, corresponding to different computational budget for the stage. For example, there are 5 branches for the stage 1 in Figure 2, the numbers of blocks for these 5 branches are, respectively, 1, 2, 3, 4, and 5. We consider the whole network as a supernet $T = \{T_1, T_2, T_3, T_4\}$, where $T_i$ at the $i$th stage has $K_i$ branches, i.e. $T_i = \{t_i^k | k = 1...K_i\}$. Then an allocation strategy can be represented as $\tau = [\tau_1, \tau_2, \tau_3, \tau_4]$, where $\tau_i$ denote the number of blocks in the $i$th branch. All blocks in the same stage have the same structure. $\sum_{i=1}^{4} \tau_i = N$ for a network with $N$ blocks. For example, the original ResNet101 has $\tau = [3, 4, 23, 3]$ and $N = 33$

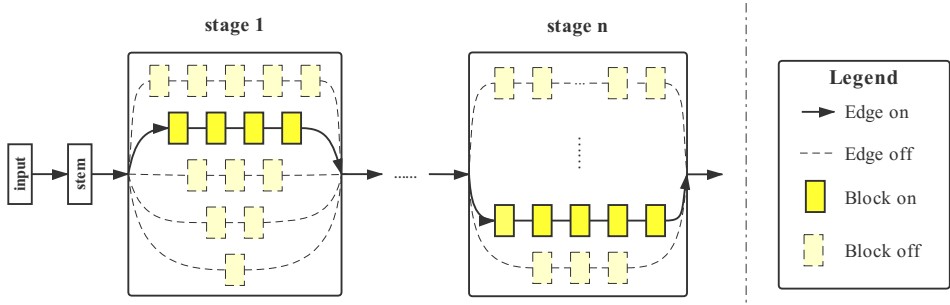

Figure 2: Stage reallocation in different resolution. In supernet training, we random sample a choice in each stage and optimize corresponding weights. In reallocation searching, eligible strategies are evaluated according to the computation budget.

residual blocks. We make a constraint that each stage at least has one convolutional block. We would like to find the best allocation strategy of ResNet101 is among the $\binom{32}{3}$ possible choices. Since validating a single detection architecture requires hundreds of GPU-hours, it not realist to find the optimal architecture by human trails.

On the other hand, we would like to learn stage reallocation strategy for different computation budgets simultaneously. Different applications require CNNs of different numbers of layers for achieving different latency requirements. This is why we have ReseNet18, ReseNet50, ReseNet101, etc. We build a search space to cover all the candidate instances in a certain series, e.g. ResNet series. After considering the trade off between granularity and range, we set the numbers of blocks for $T_1$ and $T_2$ as $\{1, 2, 3, 4, 5, 6, 7, 8, 9, 10\}$, and set the numbers of blocks for $T_3$ as $\{2, 3, 5, 6, 9, 11, 14, 17, 20, 23\}$, for $T_4$ as $\{2, 3, 4, 6, 7, 9, 11, 13, 15, 17\}$ for the ResNet series. The stage reallocation space of MobileNetV2 (Sandler et al., 2018) and ResNeXt (Xie et al., 2017) can be found in Appendix A.2.

### 3.2.2 CONVOLUTION REALLOCATION SPACE

To reallocate the computation across spatial position, we utilize dilated convolution Li et al. (2019), Li et al. (2018). Dilated convolution effects the ERF by performing convolution at sparsely sampled locations. Another good feature of dilated convolution is that dilation introduce no extra parameter and computation. We define a choice block to be a basic unit which consists of multiple dilations and search for the best computation allocation. For ResNet Bottleneck, we modify the center $3 \times 3$ convolution. For ResNet BasicBlock, we only modify the second $3 \times 3$ convolution to reduce search space and searching time. We have three candidates in our operation set $\mathcal{O}$: $\{dilated\ convolution\ 3 \times 3\ with\ dilation\ rate\ i | i = 1, 2, 3\}$. Across the entire ResNet50 search space, there are therefore $3^{16} \approx 4 \times 10^7$ possible architectures.

### 3.3 HIERARCHICAL SEARCH FOR OBJECT DETECTION

We propose a hierarchical search procedure to cope with the complex reallocation space. Firstly, the stage space is explored to find the best computation allocation for different resolution. Then, the operation space is explored to further improve the architecture with better spatial allocation.

### 3.3.1 STAGE REALLOCATION SEARCH

To reduce the side effect of weights coupling, we adopt the uniform sampling in supernet training(a.k.a single-path one-shot) (Guo et al., 2019). After the supernet training, we can validate the allocation strategies $\tau \in T$ directly on the task detection task. Model accuracy(COCO AP) is defined as $AP_{val}(\mathcal{N}(\tau, w))$. We set the block number constraint $N$. We can find the best allocation strategy in the following equation:

$$\tau^* = \underset{\sum_{i=1}^{4} \tau_i = N}{\arg\max}\ AP_{val}(\mathcal{N}(\tau, w)). \tag{1}$$

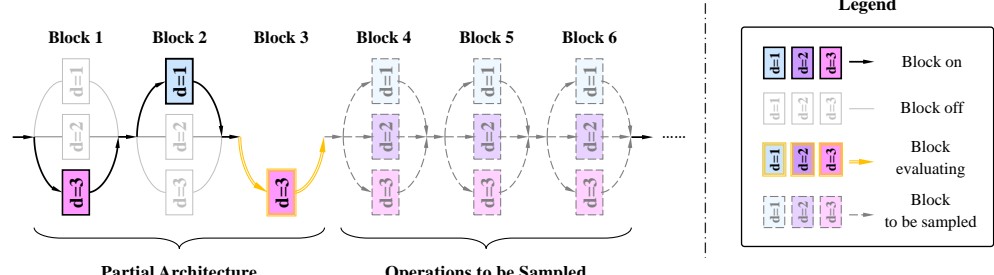

Figure 3: Evaluation of a choice in block operation search approach. As shown in figure, we have partial architecture of block 1 and block 2, and now we need to evaluate the performance of convolution with dilated rate 3 in the third block. We uniform sample the operation of rest blocks to generate a temporary architecture and then evaluate the choice through several temporary architectures.

### 3.3.2 BLOCK OPERATION SEARCH

---

**Algorithm 1:** Greedy operation search algorithm

---

**Input:** Number of blocks $B$; Possible operations set of each blocks $O = \{O_i \mid i = 1, 2, ..., B\}$; Supernet with trained weights $\mathcal{N}(O, W^*)$; Dataset for validation $D_{val}$; Evaluation metric $AP_{val}$;.

**Output:** Best architecture $o^*$

Initialize top $K$ partial architecture $p = \varnothing$

**for** *i = 1, 2, ..., B* **do**

    $p_{extend} = p \times O_i$                                             $\triangleright \times$ denotes Cartesian product

    $result = \{(arch, AP) \mid arch \in p_{extend}, AP = evaluate(arch)\}$

    $p = choose\_topK(result)$

**end**

**Output:** Best architecture $o^* = choose\_top1(p)$.

---

By introducing the operation allocation space as in Sec. 3.2.2, we can reallocate the computation across spatial position. Same as stage reallocation search, we train an operation supernet adopting random sampling in each choice block (Guo et al., 2019). For architecture search process, previous one-shot works use random search (Brock et al., 2017; Bender et al., 2018) or evolutionary search (Guo et al., 2019). In our approach, We propose a greedy algorithm to make sequential decisions to obtain the final result. We decode network architecture $o$ as a sequential of choices $[o_1, o_2, ..., o_B]$. In each choice step, the top $K$ partial architectures are maintained to shrink the search space. We evaluate each candidate operation from the first choice block to the last. The greedy operation search algorithm is shown in Algorithm 1.

The hyper-parameter $K$ is set equal to 3 in our experiment. We first extend the partial architecture in the first block choice which contains three partial architectures in $p_{extend}$. Then we expand the top 3 partial architectures into the whole length $B$, which means that there are $3 \times 3 = 9$ partial architectures in other block choice. For a specific partial architecture $arch$, we sample the operation of the unselected blocks uniformly for $c$ architectures where $c$ denotes mini batch number of $D_{val}$. We validate each architecture on a mini batch and combine the results to generate $evaluate(arch)$. We finally choose the best architecture to obtain $o^*$.

## 4 EXPERIMENTS AND RESULTS

### 4.1 DATASET AND IMPLEMENTATION DETAILS

**Dataset** We evaluate our method on the challenging MS COCO benchmark (Lin et al., 2014). We split the 135K training images *trainval135* into 130K images *archtrain* and 5K images *archval*. First, we train the supernet using *archtrain* and evaluate the architecture using *archval*. After the architecture is obtained, we follow other standard detectors (Ren et al., 2015; Lin et al., 2017a) on using ImageNet (Russakovsky et al., 2015) for pre-training the weights of this architecture. The final model

Table 1: Faster RCNN + FPN detection performance on COCO *minival* for different backbones using our computation reallocation (denoted by 'CR-x'). FLOPs are measured on the whole detector(w/o ROIAlign layer) using the input size $800 \times 1088$, which is the median of the input size on COCO.

| Backbone | FLOPs(G) | AP | $AP_{50}$ | $AP_{75}$ | $AP_s$ | $AP_m$ | $AP_l$ |
|----------|----------|-----|-----------|-----------|--------|--------|--------|
| MobileNetV2 | 121.1 | 32.2 | 54.0 | 33.6 | 18.1 | 34.9 | 42.1 |
| CR-MobileNetV2 | 121.4 | **33.9** | **56.2** | **35.6** | **19.7** | **36.8** | **44.8** |
| ResNet18 | 147.7 | 32.1 | 53.5 | 33.7 | 17.4 | 34.6 | 41.9 |
| CR-ResNet18 | 147.6 | **33.8** | **55.8** | **35.4** | **18.2** | **36.2** | **45.8** |
| ResNet50 | 192.5 | 36.4 | 58.6 | 38.7 | 21.8 | 39.7 | 47.2 |
| CR-ResNet50 | 192.7 | **38.3** | **61.1** | **40.9** | **21.8** | **41.6** | **50.7** |
| ResNet101 | 257.3 | 38.6 | 60.7 | 41.7 | **22.8** | 42.8 | 49.6 |
| CR-ResNet101 | 257.5 | **40.2** | **62.7** | **43.0** | 22.7 | **43.9** | **54.2** |

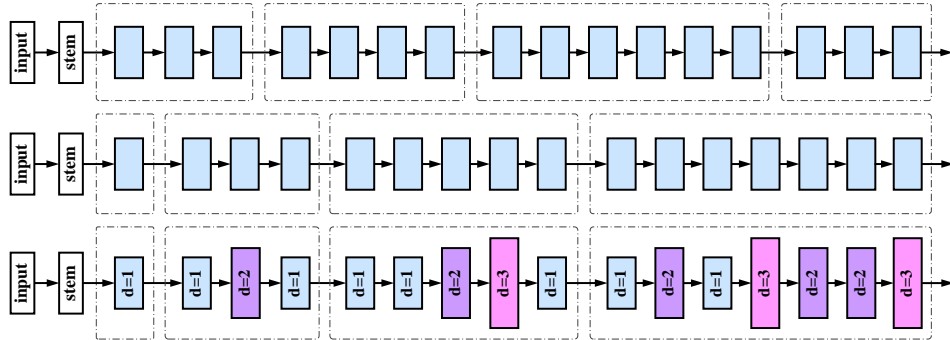

Figure 4: Architecture sketches. From top to bottom, they are baseline ResNet50, stage reallocation SCR-ResNet50 and final CR-ResNet50.

is fine-tuned on the whole COCO *trainval135* and validated on COCO *minival*. Another detection dataset VOC (Everingham et al., 2015) is also used. We use VOC $trainval2007+trainval2012$ as our training dataset and VOC $test2007$ as our vaildation dataset.

**Implementation details** The supernet training setting details can be found in Appendix A.1. For the training of our searched models, the input images are resized to have a short side of 800 pixels or a long side of 1333 pixels. We use stochastic gradient descent (SGD) as optimizer with 0.9 momentum and 0.0001 weight decay. For fair comparison, all our models are trained for 13 epochs, known as $1\times$ schedule (Girshick et al., 2018). We use multi-GPU training over 8 1080TI GPUs with total batch size 16. The initial learning rate is 0.00125 per image and is divided by 10 at 8 and 11 epochs. Warm-up and synchronized BatchNorm (SyncBN) (Peng et al., 2018) are adopted for both baselines and our searched models.

## 4.2 MAIN RESULTS

### 4.2.1 COMPUTATION REALLOCATION PERFORMANCE

We denote the architecture using our computation reallocation by prefix 'CR-', e.g. CR-ResNet50. Our final architectures have the almost the same FLOPs as the original network(the negligible difference in FLOPs is from the BatchNorm layer and activation layer). As shown in Table 1, our CR-ResNet50 and CR-ResNet101 outperforms the baseline by 1.9% and 1.6% respectively. It is worth mentioning that many mile-stone backbone improvements also only has around 1.5% gain. For example, the gain is 1.5% from ResNet50 to ResNeXt50-32x4d as indicated in Table 4. In addition, we run the baselines and searched models under longer $2\times$ setting (results shown in Appendix A.4). It can be concluded that the improvement from our approach is consistent.

Table 2: Faster RCNN + FPN detection performance on VOC $test2007$. Our computation realloca-tion models are denoted by 'CR-x'

|  | ResNet50 | CR-ResNet50 | ResNet101 | CR-ResNet101 |
|---|---|---|---|---|
| $AP_{50}$ | 84.1 | **85.1** | 85.8 | **86.5** |

Table 3: Mask RCNN detection and instance segmentation performance on COCO *minival* for dif-ferent backbones using our computation reallocation (denoted by 'CR-x'). Box and Seg are the AP (%) of the bounding box and segmentation results respectively.

| Backbone | FLOPs | Seg | $Seg_s$ | $Seg_m$ | $Seg_l$ | Box | $Box_s$ | $Box_m$ | $Box_l$ |
|---|---|---|---|---|---|---|---|---|---|
| MobileNetV2 | 189.5 | 30.6 | 15.3 | 33.2 | 42.2 | 33.1 | 18.8 | 35.8 | 43.3 |
| CR-MobileNetV2 | 189.8 | **31.8** | **16.3** | **34.3** | **44.1** | **34.6** | **19.9** | **37.3** | **45.7** |
| ResNet50 | 261.2 | 33.9 | 17.4 | 37.3 | 46.6 | 37.6 | 21.8 | 41.2 | 48.9 |
| CR-ResNet50 | 261.0 | **35.2** | **17.6** | **38.5** | **49.4** | **39.1** | **22.2** | **42.3** | **52.3** |
| ResNet101 | 325.9 | 35.6 | 18.6 | 39.2 | 49.5 | 39.7 | 23.4 | 43.9 | 51.7 |
| CR-ResNet101 | 325.8 | **36.7** | **19.4** | **40.0** | **52.0** | **41.5** | **24.2** | **45.2** | **55.7** |

Our CR-ResNet50 and CR-ResNet101 are especially effective for large objects(3.5%, 4.8% im-provement for $AP_l$). To understand these improvements, we depict the architecture sketches in Figure 4. We can find in the stage-level, our Stage CR-ResNet50 reallocate more capacity in deep stage. It reveals the fact that the budget in shallow stage is redundant while the resources in deep stage is limited. This pattern is consistent with ERF as in Figure 1. In operation-level, dilated con-volution with large rates tends to appear in the deep stage. We explain the shallow stage needs more dense sampling to gather exact information while deep stage aims to recognize large object by more sparse sampling. The dilated convolutions in deep stage further explore the network potential to detect large objects, it is an adaptive way to balance the ERF. For light backbone, our CR-ResNet18 and CR-MobileNetV2 both improves 1.7% AP over the baselines with all-round $AP_s$ to $AP_l$ im-provements. For light network, it is a more efficient way to allocate the limited capacity in the deep stage for the discriminative feature captured in the deep stage can benefit the shallow small object by the FPN top-down pathway.

### 4.2.2 TRANSFERABILITY VERIFICATION

**Different dataset** We transfer our searched model to another object detection dataset VOC (Ev-eringham et al., 2015). Training details can be found in Appendix A.3. We denote the VOC metric mAP@0.5 as $AP_{50}$ for consistency. As shown in Table 2, our CR-ResNet50 and CR-ResNet101 achieves $AP_{50}$ improvement 1.0% and 0.7% comparing with the already high baseline.

**Different task** Segmentation is another task that is highly sensitive to the ERF (Hamaguchi et al., 2018; Wang et al., 2018). Therefore, we transfer our computation reallocation network into the instance segmentation task by using the Mask RCNN (He et al., 2017) framework. The experimental results on COCO are shown in Table 3. The instance segmentation AP of our CR-MobileNetV2, CR-ResNet50 and CR-ResNet101 outperform the baseline respectively by 1.2%, 1.3% and 1.1% absolute AP. We also achieve bounding box AP improvement by 1.5%, 1.5% and 1.8% respectively.

**Different head/neck** Our work is orthogonal to other improvements on object detection. We exploit the SOTA detector Cascade Mask RCNN (Cai & Vasconcelos, 2018) for further verification. The detector equipped with our CR-Res101 can achieve 44.5% AP, better than the regular Res101 43.3% baseline by a significant 1.2% gain. Additionally, we evaluate replacing the original FPN with a searched NAS-FPN (Ghiasi et al., 2019) neck to strength our results. The Res50 with NAS-FPN neck can achieve 39.6% AP while our CR-Res50 with NAS-FPN can achieve 41.0% AP using the same $1\times$ setting. More detailed results can be found in Appendix A.4.

Table 4: COCO *minival* AP (%) evaluating stage reallocation performance for different networks. Res50 denotes ResNet50, similarly for Res101. ReX50 denotes ResNeXt50, similarly for ReXt101.

|             | MbileNetV2 | Res18 | Res50 | Res101 | ReX50-32×4d | ReX101-32×4d |
|-------------|------------|-------|-------|--------|-------------|--------------|
| Baseline AP | 32.2       | 32.1  | 36.4  | 38.6   | 37.9        | 40.6         |
| Stage-CR AP | **33.5**   | **33.4** | **37.4** | **39.5** | **38.9**  | **41.5**     |

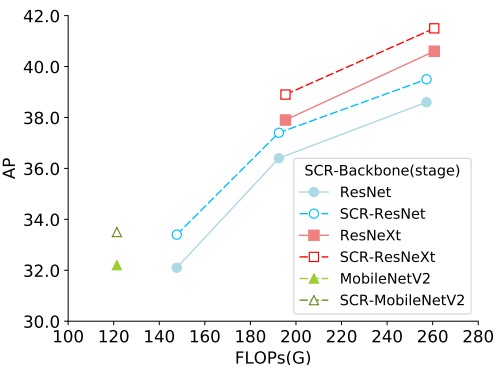

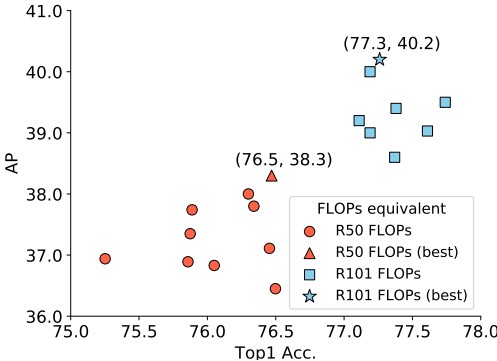

Figure 5: Detector FLOPs(G) versus AP on COCO *minival*. The bold lines and dotted lines are the baselines and our stage computation real-location models(SCR-) respectively.

Figure 6: Top1 accuracy on ImageNet validation set versus AP on COCO *minival*. Each dot is a model which has equivalent FLOPs as the base-line.

### 4.3 ANALYSIS

#### 4.3.1 EFFECT OF STAGE REALLOCATION

Our design includes two parts, stage reallocation search and block operation search. In this section, we analyse the effectiveness of stage reallocation search alone. Table 4 shows the performance comparison between the baseline and the baseline with our stage reallocation search. From light MobileNetV2 model to heavy ResNeXt101, our stage reallocation brings a solid average 1.0% AP improvement. Figure 5 shows that our Stage-CR network series yield overall improvements over baselines with negligible difference in computation. The stage reallocation results for more models are shown in Appendix A.2. There is a trend to reallocate the computation from shallow stage to deep stage. The intuitive explanation is that reallocating more capacity in deep stage results in a balanced ERF as Figure 1 shows and can enhance the ability to detect medium and large object.

#### 4.3.2 CORRELATIONS BETWEEN CLS. AND DET. PERFORMANCE

Often, a large AP increase could be obtained by simply replacing backbone with stronger network, e.g. from ResNet50 to ResNet101 and then to ResNeXt101. The assumption is that strong network can perform well on both classification and detection tasks. We further explore the performance correlation between these two tasks by a lot of experiments. We draw ImageNet top1 accuracy versus COCO AP correlation in Figure 6 for different architectures of the same FLOPS. Each dot is a single network architecture. We can easily find that although the performance correlation between these two tasks is basically positive, better classification accuracy may not always lead to better detection accuracy. This study further shows the gap between these two tasks.

## 5 CONCLUSION

In this paper, we present CR-NAS (Computation Reallocation Neural Architecture Search) that can learn computation reallocation strategies across different resolution and spatial position. We design

a two-level reallocation space and a novel hierarchical search procedure to cope with the complex search space. Extensive experiments show the effectiveness of our approach. The discovered model has great transfer-ability to other detection neck/head, other dataset and other vision tasks. Our CR-NAS can be used as a plugin to other detection backbones to further booster the performance under certain computation resources.

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

## A  APPENDIX

### A.1  SUPERNET TRAINING

Both stage and operation supernets use exactly the same setting. The supernet training process adopt the 'pre-training and fine-tuning' paradigm. For ResNet and ResNeXt, the supernet channel distribution is $[32, 64, 128, 256]$.

**Supernet pre-training.** We use ImageNet-1k for supernet pre-training. We use stochastic gradient descent (SGD) as optimizer with 0.9 momentum and 0.0001 weight decay. The supnet are trained for 150 epochs with the batch size 1024. To smooth the jittering in the training process, we adopt the cosine learning rate decay (Loshchilov & Hutter, 2016) with the initial learning rate 0.4. Warming up and synchronized-BN (Peng et al., 2018) are adopted to help convergence.

**Supernet fine-tuning.** We fine tune the pretrained supernet on *archtrain*. The input images are resized to have a short side of 800 pixels or a long side of 1333 pixels. We use stochastic gradient descent (SGD) as optimizer with 0.9 momentum and 0.0001 weight decay. Supernet is trained for 25 epochs (known as $2\times$ schedule (Girshick et al., 2018)). We use multi-GPU training over 8 1080TI GPUs with total batch size 16. The initial learning rate is 0.00125 per image and is divided by 10 at 16 and 22 epochs. Warm-up and synchronized BatchNorm (SyncBN) (Peng et al., 2018) are adopted to help convergence.

### A.2  REALLOCATION SETTINGS AND RESULTS

**stage allocation space** For ResNeXt, the stage allocation space is exactly the same as ResNet series. For MobileNetV2, original block numbers in Sandler et al. (2018) is defined by n=$[1, 1, 2, 3, 4, 3, 3, 1, 1, 1]$. We build our allocation space on the the bottleneck operator by fixing stem and tail components. A architecture is represented as $m = [1, 1, m_1, m_2, m_3, m_4, m_5, 1, 1, 1]$. The allocation space is $M = [M_1, M_2, M_3, M_4, M_5]$. $M_1, M_2 = \{1, 2, 3, 4, 5\}$, $M_3 = \{3, 4, 5, 6, 7\}$, $M_4, M_5 = \{2, 3, 4, 5, 6\}$. It's worth to mention the computation cost in different stage of $m$ is not exactly the same because of the abnormal channels. We format the weight as $[1.5, 1, 1, 0.75, 1.25]$ for $[m_1, m_2, m_3, m_4, m_5]$.

**computation reallocation results** We propose our CR-NAS in a sequential way. At first we reallocate the computation across different resolution. The Stage CR results is shown in Table A.2

Table 5: Stage reallocation strategies of different networks. MV2 denotes MobileNetV2. Res18 denotes ResNet18, similarly for Res50, Res101. ReX50 denotes ResNeXt50-32×4d, similarly for ReXt101.

|  | MV2 | Res18 | Res50 | Res101 | ReX50 | ReX101 |
|---|---|---|---|---|---|---|
| Baseline | [1,1,2,3,4,3,3,1,1,1] | [2,2,2,2] | [3,4,6,3] | [3,4,23,3] | [3,4,6,3] | [3,4,23,3] |
| Stage CR | [1,1,2,2,3,4,4,1,1,1] | [1,1,2,4] | [1,3,5,7] | [2,3,17,11] | [2,2,6,6] | [3,4,15,11] |

Then we search for the spatial allocation by adopting the dilated convolution with different rates. the operation code as. we denote our final model as

Table 6: Final network architectures.

|  | stage code | operation code |
|---|---|---|
| CR-MobileNetV2 | [1,1,2,2,3,4,4,1,1,1] | [0, 1, 0, 1, 0, 2, 0, 1, 1, 0, 0, 1, 1, 0, 1, 1, 0, 2, 0, 0] |
| CR-ResNet18 | [1,1,2,4] | [0, 0, 1, 0, 1, 0, 2, 1] |
| CR-ResNet50 | [1,3,5,7] | [0, 0, 1, 0, 0, 0, 1, 2, 0, 0, 1, 0, 2, 1, 1, 2] |
| CR-ResNet101 | [2,3,17,11] | [0, 0, 0, 0, 0, 0, 1, 0, 1, 0, 0, 0, 2, 0, 0, 0, 1 , 0, 1, 0, 1, 0, 1, 1, 0, 0, 1, 0, 1, 2, 0, 1, 1] |

[0 ] dilated conv with rate 1(normal conv) [1 ] dilated conv with rate 2 [2 ] dilated conv with rate 3

Our final model can be represnted as a series of allocation codes.

## A.3 IMPLEMENTATION DETAILS OF VOC

We use the VOC $trainval2007+trainval2012$ to server as our whole training set. We conduct our results on the VOC $test2007$. The pretrained model is apoted. The input images are resized to have a short side of 600 pixels or a long side of 1000 pixels. We use stochastic gradient descent (SGD) as optimizer with 0.9 momentum and 0.0001 weight decay. We train for 18 whole epochs for all models. We use multi-GPU training over 8 1080TI GPUs with total batch size 16. The initial learning rate is 0.00125 per image and is divided by 10 at 15 and 17 epochs. Warm-up and synchronized BatchNorm (SyncBN) (Peng et al., 2018) are adopted to help convergence.

## A.4 MORE EXPERIMENTS

**longer schedule** $2\times$ schedule means training totally 25 epochs as indicated in Girshick et al. (2018). The initial learning rate is 0.00125 per image and is divided by 10 at 16 and 22 epochs. Other training settings is exactly the same as in $1\times$.

Table 7: Longer $2\times$ Faster RCNN + FPN detection performance on COCO *minival* for different backbones using our computation reallocation (denoted by 'CR-x').

| Backbone | FLOPs (G) | AP | $AP_{50}$ | $AP_{75}$ | $AP_s$ | $AP_m$ | $AP_l$ |
|---|---|---|---|---|---|---|---|
| ResNet50 | 192.5 | 37.6 | 59.5 | 40.6 | **22.4** | 40.8 | 48.5 |
| CR-ResNet50 | 192.7 | **39.3** | **60.8** | **42.1** | 22.0 | **42.4** | **52.5** |
| ResNet101 | 257.3 | 39.8 | 61.5 | 43.2 | 23.2 | 44.0 | 51.4 |
| CR-ResNet101 | 257.5 | **41.2** | **62.5** | **44.6** | **23.7** | **44.8** | **54.6** |

**Powerful detector** The Cascade Mask RCNN (Cai & Vasconcelos, 2018) is a SOTA multi-stage object detector. The detector is trained for 20 epochs. The initial learning rate is 0.00125 per image and is divided by 10 at 16 and 19 epochs. Warming up and synchronized-BN (Peng et al., 2018) are adopted to help convergence.

Table 8: SOTA Cascade Mask RCNN detection performance on COCO *minival* for ResNet101 and our CR-ResNet101.

| Detector | Bakebone | AP | AP50 | AP75 | APs | APm | APl |
|---|---|---|---|---|---|---|---|
| Cascade Mask | Res101 | 43.3 | 61.5 | 47.3 | 24.7 | 46.6 | 57.6 |
| Cascade Mask | CR-Res101 | **44.5** | **62.6** | **48.0** | **25.6** | **47.7** | **60.2** |

**Powerful searched neck** NAS-FPN (Ghiasi et al., 2019) is a powerful scalable feature pyramid architecture searched for object detection. We reimplement NAS-FPN (7 @ 384) in Faster RCNN (The original paper is implemented in RetinaNet (Lin et al., 2017b)). The detector is training under $1\times$ setting as described in 4.1.

Table 9: Faster RCNN + NAS-FPN detection performance on COCO *minival* for ResNet50 and our CR-ResNet50.

| Bakebone | Neck | AP | AP50 | AP75 | APs | APm | APl |
|---|---|---|---|---|---|---|---|
| Res50 | NAS-FPN (7 @ 384) | 39.6 | 60.4 | 43.3 | **22.8** | 42.8 | 51.5 |
| CR-Res50 | NAS-FPN (7 @ 384) | **41.0** | **61.2** | **44.2** | 22.7 | **44.9** | **54.2** |

