# OpenReview forum: "Computation Reallocation for Object Detection"
_ICLR.cc/2020/Conference — Accept (Poster)_

### Official Review · AnonReviewer2 · 2019-10-15
**Official Blind Review #671**

**Rating:** 3

**Review:**

<Strengths>
+ This paper performs architecture search for object detection, especially for the computation allocation across different resolutions. It is a new application for NAS research.
+ The proposed approach shows some marginal improvement of object detection accuracy across multiple backbones and datasets.
+ This work proposes a new formulation to apply NAS approach to object detectors, including linking between the ERF and the computation allocation of backbone and two-level hierarchical search for stages and convolution operations.

<Weakness>
1. This paper can be regarded as an engineering work for a new domain of problem with little technical novelty.
- From the perspective of NAS research, the proposed approach has little technical novelty; instead, it seems like an application of existing techniques (or even simpler ones) (e.g.  to a new domain of problem - object detection.
- Given that the NAS is only applied to CNN backbones in this paper, the novelty (of proposing a new task) may be further weakened.
- The novelty of this work over existing works of “NAS on detection” (Chen et al 2019, Wang et al 2019 and Ghiasi et al 2019) is not justified.

2. Experimental results are rather weak.
- This paper only focuses on showing that the proposed method marginally improves the performance of basic backbones (MobileNet V2 and ResNet 18/50/101). However, the improvement gaps are rather marginal (about 1.0% in average as shown in Fig.5).
- This paper does not compare its performance with other SOTA detection methods but only compare with some baselines instead. However, the performances of baselines are too low; for example, in Table 1, the mAP of the ResNet101 baseline is 38.6, which is lower by about 10 than the SOTA detector with the same ResNet101+FRCNN+FPN. It may not be convincing to use the baselines that have more than 20% lower accuracy compared to SOTA and show only ~1.x improvement over it.
- Given that the proposed approach is applicable to any backbones and detectors, more experiments should be done using recent stronger baselines (including many recent tricks like RoIAlign and DCN).
- Another experimental weakness is lack of comparison with existing NAS methods on object detections such as (Chen et al 2019, Wang et al 2019 and Ghiasi et al 2019). Even though experiment settings here may be different from those of these papers,  they should be compared in any reasonable ways. Moreover, the detection accuracies reported in this paper are not as good as the numbers of these papers. This paper argues that the proposed approach is complementary to these methods, so it would be good to report the results of the combined model with them.

3. The paper is written poorly.
- There are many grammatically wrong and awkward expressions. The draft should be thoroughly proofread.

<Conclusion>
Although this work shows a new application of NAS for object detection, my initial decision is ‘weak reject’ mainly due to lack of technical novelty, limited experiments and poor writing.

<Post-rebuttal comments>
Authors' responses partly resolved my concerns on the experiments. I have no object to accept this paper.

**Experience Assessment:**

I have published one or two papers in this area.

**Review Assessment: Checking Correctness Of Derivations And Theory:**

I assessed the sensibility of the derivations and theory.

**Review Assessment: Checking Correctness Of Experiments:**

I carefully checked the experiments.

**Review Assessment: Thoroughness In Paper Reading:**

I read the paper thoroughly.

---

> ### Author Response · Authors · 2019-11-12
> **Response to Reviewer #2**
>
> We sincerely thank you for the detailed feedback. We will explain your concerns point by point.
>
> Q1: From the perspective of NAS research, the proposed approach has little technical novelty; instead, it seems like an application of existing techniques (or even simpler ones) (e.g. to a new domain of problem object detection.
>
> A1: NAS works only search for operation in the certain layer, our work is different from them by searching for the computation allocation across different resolution. Computation allocation across feature resolutions is an obvious issue that has not been studied by NAS. Simply treating the computation allocation as operation is not appropriate, we carefully design a search space that facilitates the use of existing search for finding good solution, which is non-trivial and novel. Typically, our stage search space can cover all the candidate instances in a certain series, e.g. ResNet series. The reusable search space can further reduce the searching cost and adapt different computational requirements. Details can be found in 3.2.1.
>
> Q2: Given that the NAS is only applied to CNN backbones in this paper, the novelty (of proposing a new task) may be further weakened.
>
> A2: Object detection is a fundamental computer vision topic with wide application and with many papers published in ICLR, NeurIPS, etc. It is a fundamental topic to search the proper CNN backbones for specific applications like object detection. To our best knowledge, there is no such a reallocation work yet. Following our clues, we can reallocate the computation resources among different detector components, e.g. head, neck. That is a fascinating future work of our approach.
>
> Q3: The novelty of this work over existing works of “NAS on detection” (Chen et al 2019, Wang et al 2019 and Ghiasi et al 2019) is not justified.
>
> A3: NAS-FPN(Ghiasi et al 2019) and NAS-FCOS(Wang et al 2019) fix the architecture of backbone CNN and search for the neck or head of detector. The search space of Wang et al 2019 is directly inherited from the classification task which is suboptimal for object detection task. Our work is orthogonal to these approaches by searching for the computation allocation pattern.
>
> Q4: This paper only focuses on showing that the proposed method marginally improves the performance of basic backbones ... the improvement gaps are rather marginal (about 1.0% in average as shown in Fig.5).
>
> A4: Figure 5 only shows our stage reallocation improvements over the baselines. Our final searched models outperform the baseline with a significant 1.6% to 1.9% gain without any additional computation budget as indicated in Table 1. Many mile-stone backbone improvements also only has around 1.5% gain. For example, the gain is 1.5% from ResNet50 to ResNeXt50-32x4d as indicated in Table 4.
>
> Q5: This paper does not compare its performance with other SOTA detection methods but only compare with some baselines ... more experiments should be done using recent stronger baselines.
>
> A5: We claim that the effectiveness of our approach comes from the relative gain under exactly the same setting. We exploit the SOTA detector Cascade Mask RCNN for further verification. The detector equipped with our CR-Res101 can achieve 44.5% AP, better than the  regular Res101 43.3% baseline by a significant 1.2% gain. It is worth mentioning that the improvement by replacing the Res50 with Res101 is nearly 1.0% for Cascade Mask RCNN as in [1].
>
> Detector             Bakebone      AP              AP50      AP75     APs       APm      APl
> Cascade Mask   Res101          43.3             61.5       47.3      24.7       46.6       57.6
> Cascade Mask   CR-Res101    44.5(+1.2)   62.6       48.0      25.6       47.7       60.2
>
> Q6: Another experimental weakness is lack of comparison with existing NAS methods on object detections ... good to report the results of the combined model with them.
>
> A6:  Our approach is complementary to existing NAS methods on object detection. We evaluate replacing the original FPN with a searched NAS-FPN[1] neck to strength our results. The Res50 with NAS-FPN neck can achieve 39.6% AP while our CR-Res50 with NAS-FPN can achieve 41.0% AP using the same '1x' setting.
>
> Backbone     Neck                            AP              AP50      AP75     APs       APm      APl
> Res50            NAS-FPN (7@384)     39.6            60.4       43.3       22.8      42.8       51.5
> CR-Res50      NAS-FPN (7@384)     41.0(+1.4)  61.2       44.2       22.7      44.9       54.2
>
> Q7: There are many grammatically wrong and awkward expressions. The draft should be thoroughly proofread.
>
> A7: Thank you for your precious advice. We will thoroughly proofread the paper. We have tried our best to fix the typos as below.
>
> 1. is take directly -> is taken directly
> 2. Dilated convolution effects the ERF -> Dilated convolution affects the ERF
> 3. ResNet Bottlneck -> ResNet Bottleneck
>
> [1] https://github.com/open-mmlab/mmdetection/blob/master/docs/MODEL_ZOO.md

---

### Official Review · AnonReviewer1 · 2019-10-30
**Official Blind Review #1**

**Rating:** 6

**Review:**

The paper attempts to apply neural architecture search (NAS) to re-arrange, or re-allocate the network backbone blocks and the convolution filters for object detection. The search space is two-fold: 1) the network is allowed to search over allocation of different number of blocks in the backbone (e.g. ResNet, MobileNet); 2) the network is allowed to choose the dilation of each of the block. A one-shot NAS method is adopted for efficient search. After search, the model is shown to have 1) better AP results; and 2) more balanced effective receptive field (ERF).

+ I am not aware of any work that performs search on backbone architectures for object detection yet. So the idea itself is novel;
+ The visualization of the ERF is interesting -- it reals that ERF is more balanced after searching.

- My biggest concern is in results. It seems for Faster R-CNN with FPN, the detection results should be higher in general (e.g. R-50-FPN should at least give ~37 Ap with 1x training, and can reach 38 if it trains longer --  the same as CR-R-50-FPN in Table 1). Therefore I am not fully convinced that the searched results are obtaining meaningful gains -- maybe a result that trains longer can help here.
- Related -- I think while the idea is interesting, the limited improvement is hurting the significance of the work. In fact, to me the most important result would be on Fig 5, where it compares the speed/accuracy trade-off (directly comparing accuracy is meaning less unless the paper reaches state-of-the-art -- which is around 50 now); however, again no significant gains. Here it is because many "improvements" have been proposed after Faster/Mask R-CNN as baseline.
- (Minor) I am not sure the computation of possible choices 33^3 is accurate for the search space, because some of these 33^3 blocks are identical and therefore redundant.
- 4.3.2 is a bit misleading. At least improving backbone helps improves object detection performance (as far as I know), the plot also shows quite a bit of correlation between classification and detection performance -- please report at least the correlation for the points on the figure.

Question:
* From Fig 4, it seems for baseline R-50, it would be best to allocate more computation on the later stages (e.g. 1st one only has 3, and last has 7), Is it true for other search results? Is it true for other head (instead of FPN)? Are there intuitive explanations for that?
* Also from Fig 4, it seems the network tend to favor more dilated convolutions toward the end? Does it have something to do with the ERF balancing?

Despite the concerns, I am still in favor of accepting the paper, as the paper concerns both accuracy and speed for detection, and applying NAS to such kind of search reals some interesting patterns (please answer the questions above).

**Experience Assessment:**

I have published one or two papers in this area.

**Review Assessment: Checking Correctness Of Derivations And Theory:**

N/A

**Review Assessment: Checking Correctness Of Experiments:**

I assessed the sensibility of the experiments.

**Review Assessment: Thoroughness In Paper Reading:**

I read the paper at least twice and used my best judgement in assessing the paper.

---

> ### Author Response · Authors · 2019-11-12
> **Response to Reviewer #1 (Part 1)**
>
> We sincerely thank you for your comprehensive comments and constructive advices. We will explain your concerns point by point.
>
> Q1: The detection results should be higher in general ... a result that trains longer can help here.
>
> A1: We use a Pytorch implementation similar with MMDetection [1], whose '1x' ResNet50 baseline accuracy (36.4%) is the same as ours (36.4%) and is a little bit lower than the caffe2 implementation Detectron[2]. All our detectors only run once under ‘1x’ to be fair. In addition, we run the baselines and searched models under longer '2x' setting (results shown below). It can be concluded that the improvement from our approach is consistent.
>
> Detector    Train scheduler    Bakebone      AP              AP50      AP75     APs       APm      APl
> Faster         2x                           Res50            37.6             59.5       40.6       22.4       40.8      48.5
> Faster         2x                           CR-Res50      39.3(+1.6)   60.8       42.1       22.0       42.4      52.5
> Faster         2x                           Res101          39.8             61.5       43.2       23.2       44.0      51.4
> Faster         2x                           CR-Res101    41.2(+1.4)   62.5       44.6       23.7       44.8      54.6
>
> Q2: ... directly comparing accuracy is meaning less unless the paper reaches state-of-the-art ... no significant gains.
>
> A2: Figure 5 shows our stage reallocation improvements over the baselines. From light MobileNetV2 to heavy ResNeXt101, our stage reallocation brings a solid average 1.0% AP improvement over the baselines. Our final searched models outperform the baselines with a significant 1.6% to 1.9% gain without any additional computation budget(main results in Table 1). Many mile-stone backbone improvements also only has around 1.5% gain. For example, the gain is 1.5% from ResNet50 to ResNeXt50-32x4d as indicated in Table 4.
>
> We exploit the SOTA detector Cascade Mask RCNN[3] for further verification. The detector equipped with our CR-Res101 can achieve 44.5% AP, better than the  regular Res101 43.3% baseline by a significant 1.2% gain. It is worth mentioning that the improvement by replacing the Res50 with Res101 is nearly 1.0% for Cascade Mask RCNN as in [1].
>
> Detector             Bakebone      AP              AP50      AP75     APs       APm      APl
> Cascade Mask   Res101          43.3             61.5       47.3      24.7       46.6       57.6
> Cascade Mask   CR-Res101    44.5(+1.2)   62.6       48.0      25.6       47.7       60.2
>
> Additionally, we evaluate replacing the original FPN with a searched NAS-FPN[4] neck to strength our results. The Res50 with NAS-FPN neck can achieve 39.6% AP while our CR-Res50 with NAS-FPN can achieve 41.0% AP using the same '1x' setting.
>
> Backbone     Neck                            AP              AP50      AP75     APs       APm      APl
> Res50            NAS-FPN (7@384)     39.6            60.4       43.3       22.8      42.8       51.5
> CR-Res50      NAS-FPN (7@384)     41.0(+1.4)  61.2       44.2       22.7      44.9       54.2
>
> In summary, our approach can bring stable improvements over object detection tasks without any additional computation budget. All those results will be added to our final version.
>
> Q3: I am not sure the computation of possible choices 33ˆ^3 is accurate for the search space.
>
> A3: Thank you for your careful reviews. If we simplify the search space and make a constraint that each stage at least has one convolutional block, we will reduce the search space to C(32,3) which is a more intuitive.
>
> Q4: Section 4.3.2 is a bit misleading... please report at least the correlation for the points on the figure.
>
> A4: We will clarify that stronger networks generally improve the detection performance, e.g. comparing between ResNeXt50, ResNet101 with ResNet50. The performance of classification and detection has basically positive correlation as depicted in Figure 6. But better classification accuracy may not always lead to better detection performance. We compare the ResNet50, ResNeXt-50-32x4d and our CR-ResNet50 for example. ResNeXt-50-32x4d has better classification accuracy but is weaker than our CR-ResNet50 in object detection. This study further shows the gap between these two tasks.
>
> Backbone           ImageNet Top1      AP               AP50      AP75     APs       APm      APl
> Res50                  76.18                        36.4             58.6       38.7       21.8       39.7       47.2
> ResX50-32x4d    77.56(+1.38)           37.9(+1.5)   60.5       40.7       22.4       41.6       48.5
> CR-Res50            76.53(+0.35)           38.3(+1.9)   61.0       40.9       21.8       41.6       50.7

---

> > ### Author Response · Authors · 2019-11-12
> > **Response to Reviewer #1 (Part 2)**
> >
> > Q5: From Fig 4, it seems for baseline R-50, it would be best to allocate more computation on the later stages (e.g. 1st one only has 3, and last has 7), Is it true for other search results? Is it true for other head (instead of FPN)? Are there intuitive explanations for that?
> >
> > A5: The stage reallocation results for more models are shown in Table 5 (Appendix B). From light MobileNetV2 to heavy ResNeXt101-32×4d, it is a trend to reallocate the computation from shallow stage to deep stage. The intuitive explanations are two-fold. Firstly, reallocating more capacity in deep stage results in a balanced ERF as Figure 1 shows and can enhance the ability to detect medium and large object, detailed results can be found in Table 1. Secondly, the discriminative feature captured in the deep stage can benefit the shallow small object by the FPN top-down pathway. The reallocation strategy of different neck/head may not be exactly the same as in FPN, but we infer that they might be similar. The consistent improvements with other head or neck(results in A2) reveals that the direction is right.
> >
> > Q6: From Fig 4, it seems the network tend to favor more dilated convolutions toward the end? Does it have something to do with the ERF balancing?
> >
> > A6: The spatial(dilation) reallocation results are shown in Table 6 (Appendix B). The networks do favor more dilated convolution in the deep stage. We think the dilation in deep stage further explores the network potential to detect large objects, it is an adaptive way to balance the ERF.
> >
> > [1] https://github.com/open-mmlab/mmdetection/blob/master/docs/MODEL\_ZOO.md
> >
> > [2] https://github.com/facebookresearch/Detectron/blob/master/MODEL\_ZOO.md
> >
> > [3] Cai, Zhaowei, and Nuno Vasconcelos. "Cascade R-CNN: High Quality Object Detection and Instance Segmentation." arXiv preprint arXiv:1906.09756 (2019).
> >
> > [4] Ghiasi, Golnaz, Tsung-Yi Lin, and Quoc V. Le. "Nas-fpn: Learning scalable feature pyramid architecture for object detection." Proceedings of the IEEE Conference on Computer Vision and Pattern Recognition. 2019.

---

### Official Review · AnonReviewer4 · 2019-10-31
**Official Blind Review #4**

**Rating:** 6

**Review:**

This paper works on neural architecture search for object detection. Two search directions are proposed: 1) searching the number of conv blocks at each resolution (or "stage"). 2) searching the dilations for each conv block. A greedy neighbor-based search algorithm is adopted. The results show healthy improvements among different network architectures. And the searched architecture also performs well on other tasks or datasets.

Overall it is a valid paper with reasonable ideas and decent results. I like the conclusion that the searched architecture also works well on other tasks. This can be a universal replacement of the regular Resnets if people are willing to switch. However, the results are not exciting enough. The baseline models are old and it is not surprising doing an architecture search can improve. It seems that the major improvements are from re-arranging the convolutional blocks (comparing Table. 4 and Table. 1), which is one of the most straightforward directions for architecture search. The improvements of adding dilation on earlier layers are not exciting. Also, the authors do not compare to any other neural architecture search methods, which makes the improvements less convincing.

I vote for a weak rejection for now, mainly based on the limited novelty. A more interesting improvement will be (manually) comparing the searched architecture for different tasks. E.g., will all tasks prefer more layers in deeper stages or does classification prefer more layers in the middle, and segmentation prefers more layers in the beginning. I will be happy to alter my rating if the authors show more exciting observations (not limited to the above direction).


**Experience Assessment:**

I have read many papers in this area.

**Review Assessment: Checking Correctness Of Derivations And Theory:**

I assessed the sensibility of the derivations and theory.

**Review Assessment: Checking Correctness Of Experiments:**

I carefully checked the experiments.

**Review Assessment: Thoroughness In Paper Reading:**

I read the paper thoroughly.

---

> ### Author Response · Authors · 2019-11-12
> **Response to Reviewer # 4**
>
> We sincerely thank you for the valuable comments on our paper. We will explain your concerns point by point.
>
> Q1: The baseline models are old and it is not surprising doing an architecture search can improve.
>
> A1: Our searched models outperform the baseline with a significant 1.6% to 1.9% gain without any additional computation budget as indicated in Table 1. Many mile-stone backbone improvements also only has around 1.5% gain. For example, the gain is 1.5% from ResNet50 to ResNeXt50-32x4d as indicated in Table 4.
>
> We use another SOTA detector, Cascade Mask RCNN[2], for further verification.
> The detector equipped with our CR-Res101 can achieve 44.5 AP, better than the regular Res101 43.3 baseline by a significant 1.2% gain. It is worth mentioning that the improvement by replacing the Res50 with Res101 is nearly 1.0% for Cascade Mask RCNN as in [1]. The results shows that our improvement is stable on both old and new detectors.
>
> Detector             Bakebone      AP              AP50      AP75     APs       APm      APl
> Cascade Mask   Res101          43.3             61.5       47.3      24.7       46.6       57.6
> Cascade Mask   CR-Res101    44.5(+1.2)   62.6       48.0      25.6       47.7       60.2
>
> Q2: It seems that the major improvements are from re-arranging the convolutional blocks (comparing Table. 4 and Table. 1), which is one of the most straightforward directions for architecture search. The improvements of adding dilation on earlier layers are not exciting.
>
> A2: Re-arranging the conv blocks is straightforward and brings more AP gain(nearly 1.0) than adding dilations(nearly 0.7) comparing the Table 1 and Table 4. The results reveal the effectiveness of our two-level search space, typically stage reallocation space. Computation allocation across feature resolutions is a straightforward and fundamental direction that has not been studied by NAS, while our work is the first on this direction, which is the most exciting part of this paper.
>
> Q3: The authors do not compare to any other neural architecture search methods, which makes the improvements less convincing.
>
> A3: Our work is orthogonal to other NAS approaches by searching for the computation allocation pattern for backbone. We replace the original FPN with a searched NAS-FPN[3] neck to further validate the effectiveness of our appraoch. The Res50 with NAS-FPN neck can achieve 39.6% AP while our CR-Res50 with NAS-FPN can achieve 41.0% AP using the same '1x' setting. Therefore, the improvement from our approach is complementary to existing NAS for detection.
>
> Backbone     Neck                            AP              AP50      AP75     APs       APm      APl
> Res50            NAS-FPN (7@384)     39.6            60.4       43.3       22.8      42.8       51.5
> CR-Res50      NAS-FPN (7@384)     41.0(+1.4)  61.2       44.2       22.7      44.9       54.2
>
> Q4: A more interesting improvement will be (manually) comparing the searched architecture for different tasks ... does classification prefer more layers in the middle, ...
>
> A4: There is a trend to reallocate the computation from shallow stage to deep stage. There are more detailed explanations in Reviewer #1 A5. It is an interesting idea to consider whether different task or dataset would prefer different stage reallocation strategy. We show more search results on image classification. Take ResNet50 as an example, the allocation strategies are different for the different tasks of image classification and detection. While detection task prefers more layers in the last stage([1,3,5,7]), the classification task stacks more layers in second last stage([2,2,9,3]) as shown below. We give the intuitive explanations that the larger ERF benefits the detection task(stacking more conv in the end), but may not benefit the classification because the ERF is already large enough for classification (image classification has small input image size 224, while detection has larger input image size).
>
> Model                     ImageNet Top1    COCO AP      Stage code
> Res50                      76.18                      36.4               [3,4,6,3]
> SCR-Res50-Det.     76.35(+0.17)          37.4(+1.0)     [1,3,5,7]
> SCR-Res50-Cls.      76.51(+0.33)          36.9(+0.5)     [2,2,9,3]
> * -Det. means searching stage on detection,  -Cls. means searching stage on classification.
>
> Reference:
> [1] https://github.com/open-mmlab/mmdetection/blob/master/docs/MODEL_ZOO.md
> [2] Cai, Zhaowei, and Nuno Vasconcelos. "Cascade R-CNN: High Quality Object Detection and Instance Segmentation." arXiv preprint arXiv:1906.09756 (2019).
> [3] Ghiasi, Golnaz, Tsung-Yi Lin, and Quoc V. Le. "Nas-fpn: Learning scalable feature pyramid architecture for object detection." Proceedings of the IEEE Conference on Computer Vision and Pattern Recognition. 2019.

---

### Official Review · AnonReviewer3 · 2019-10-31
**Official Blind Review #3**

**Rating:** 8

**Review:**

This paper describes a neural architecture search method for computation resources allocation across feature resolutions in object detection. A two level reallocation space is proposed for both stage and spatial reallocation. The experiment results have quite nice improvements on several standard data sets.

This is a great, well written paper overall. The design and experiment settings are well described with details. In short, this is a perfect paper that I enjoy reading.

I only have very small questions and suggestions to this paper.

The paper claimed the approach is able to reallocate the engaged computation resources in a more efficient way. If I did not missing anything, the paper only shows related experiments in figure 5 and figure 6 with corresponding descriptions in 4.3.2. I hope the author could have more details in these two figures with more analysis. Personally, I think more analysis on computational effectiveness may make the paper more attractive.

We all know that neural network training may not be very stable in some settings. One thing I am curious about in this paper is whether the output network architectures from different training are always the same. If they are not the same, can you compare the differences?

I am also curious if the author could give some intuition of the network architecture of the final best network. In other words, we want to know why the final network is better than other networks. I read the Figure 4, Table 5 and Table 6, but I really cannot understand why those networks are that 'good'. Maybe, we can find some clues by answering the last paragraph.

Following the last question, we also find out that the same NAS algorithm produces different networks on different data sets. Is it because of the data set settings, or because of the content of the data sets or because of network randomness? What are your intuitions?

A detailed question in 3.2.2. Why you only modify the second 3x3 conv in ResNest BasicBlock and only modify the center 3x3 conv in ResNet Bottleneck.

Does the hyperparameter K in 3.3.2 mater a lot (like 4 or 5)?

The 4.2.2 "transfer-ability verification" is a very nice section. Do you  train NAS on VOC or only a fixed network architecture on VOC? If you did both, what is the performance difference?

**Experience Assessment:**

I do not know much about this area.

**Review Assessment: Checking Correctness Of Derivations And Theory:**

I assessed the sensibility of the derivations and theory.

**Review Assessment: Checking Correctness Of Experiments:**

I carefully checked the experiments.

**Review Assessment: Thoroughness In Paper Reading:**

I read the paper at least twice and used my best judgement in assessing the paper.

---

> ### Author Response · Authors · 2019-11-12
> **Response to Reviewer #3**
>
> Thank you for your appreciation of our paper. We are glad to answer your questions point by point.
>
> Q1: The paper only shows related experiments in figure 5 and figure 6 with corresponding descriptions in 4.3.2. ... more analysis on computational effectiveness may make the paper more attractive.
>
> A1: Figure 5 shows our stage reallocation improvements over the baselines. From light
> MobileNetV2 to heavy ResNeXt101, our stage reallocation brings a solid average 1.0% AP improvement over the baselines.
>
> As shown in Figure 6, although the performance of classification and detection has basically positive correlation, better classification accuracy may not always lead to better detection accuracy. We compare the ResNet50, ResNeXt-50-32x4d and our CR-ResNet50 for example. ResNeXt-50-32x4d has better classification accuracy but is has worse detection accuracy than our CR-ResNet50. This study further shows the gap between these two tasks.
>
> Backbone           ImageNet Top1      AP               AP50      AP75     APs       APm      APl
> Res50                  76.18                        36.4             58.6       38.7       21.8       39.7       47.2
> ResX50-32x4d    77.56(+1.38)           37.9(+1.5)   60.5       40.7       22.4       41.6       48.5
> CR-Res50            76.53(+0.35)           38.3(+1.9)   61.0       40.9       21.8       41.6       50.7
>
> Q2: Whether the output network architectures from different training are always the same. If they are not the same, can you compare the differences?
>
> A2: We find that they may have small variation in the number of conv layers and dilated conv in deep layers. But they have the same trend(more conv layers and dilated conv in the deep stage) and almost the same accuracy.
>
> Q3: Give some intuitions of the network architecture of the final best network ... why the final network is better than other networks.
>
> A3: As indicated in Figure 4, there are two trends to get 'good' networks. Firstly, rearranging more conv layers in the deep stage. Reallocating more capacity in deep stage results in a balanced ERF as Figure 1 shows and can enhance the ability to detect medium and large object(detailed results can be found in Table 1). Second, stacking more dilated convolution in the deep stage. We infer that it is because shallow stage needs more dense sampling to gather exact texture information while deep stage aims to recognize large object by more sparse sampling. It is an adaptive way to balance the ERF.
>
> Q4: The same NAS algorithm produces different networks on different data sets. Is it because of the data set settings, or because of the content of the data sets or because of network randomness?
>
> A4: Data set settings, the content of the data sets, randomness in network and sampling altogether result in  different networks on different data sets.
>
> Q5: Why you only modify the second 3x3 conv in ResNest BasicBlock and only modify the center 3x3 conv in ResNet Bottleneck.
>
> A5: We only modify one 3x3 conv layer to reduce search space and searching time. We can modify more layers, which can potentially lead to better accuracy, but requires more searching time.
>
> Q6: Does the hyperparameter K in 3.3.2 matter a lot (like 4 or 5)?
>
> A6: Hyper-parameter K is the number of partial architectures to be maintained in greedy operation search Algorithm 1. Larger K will introduce more evaluation cost because they are linear correlated. We have tried K=1 in our approach. The searched model accuracy(38.1%) is slightly lower than our final models(38.3%, K=3) for ResNet50. We think K=3 is large enough to reach good accuracy, but larger K(like 4 or 5) will not hurt the performance.
>
> Q7: Do you train NAS on VOC or only a fixed network architecture on VOC? If you did both, what is the performance difference?
>
> A7: We do not search directly on VOC, but transfer the architectures found from COCO to VOC. We infer that they may have the similar reallocation pattern on VOC. The transfer improvements reveals that the direction is right.

---

### Author Response · Authors · 2019-11-14
**Summary of revision**

Dear reviewers, thank you for the detailed feedback and constructive advices! More experiment results and changes have been added in the latest revision. We hope we have addressed all of your concerns. To clarify what have been changed, we have made the following overview of major changes.

- Add more experiments to show the consistent improvements of our searched 'CR-' model.

Added in section 4.2.2, detailed results found in Appendix A.4. We run the baselines and searched models under longer '2x' setting (Reviewer #1 Q1), using STOA Casecade Mask RCNN (Reviewer #1 Q2, Reviewer #2 Q5, Reviewer #4 Q1), using searched powerful neck NAS-FPN (Reviewer #2 Q6, Reviewer #4 Q3). It can be concluded that the improvement from our approach is consistent.

- Add more intuitive explanations  about why our searched model is better.

Added in 4.2.1 and 4.3.1. There two trends in our reallocation pattern.  Firstly, rearranging more conv layers in the deep stage(Reviewer #1 Q5, Reviewer #3 Q3). Secondly, stacking more dilated convolution in the deep stage(Reviewer #1 Q6,). We give more intuitive explanations.

- Revise the related work to compare with other NAS methods.

Revised in 2. NAS works only search for operation in the certain layer. our work is different from
them by searching for the computation allocation across different resolution. Our approach is complementary with other NAS research(Reviewer #2 Q1).

- Modify the size of stage search space of ResNet101.

Modified in 3.2.1. We simplify the search space and make a constraint that each stage at least has one convolutional block, we will reduce the search space to C(32,3) which is a more intuitive(Reviewer #1 Q3).

- Fix some typos.

---

### Decision · Program_Chairs · 2019-12-19

**Decision:**

Accept (Poster)

**Comment:**

The submission applies architecture search to object detection architectures. The work is fairly incremental but the results are reasonable. After revision, the scores are 8, 6, 6, 3. The reviewer who gave "3" wrote after the authors' responses and revision that "Authors' responses partly resolved my concerns on the experiments. I have no object to accept this paper. [sic]". The AC recommends adopting the majority recommendation and accepting the paper.